# Parental Educational Attainment and Mental Well-Being of College Students: Diminished Returns of Blacks

**DOI:** 10.3390/brainsci8110193

**Published:** 2018-10-29

**Authors:** Shervin Assari

**Affiliations:** 1Department of Psychiatry, University of Michigan, Ann Arbor, MI 48109-2700, USA; assari@umich.edu; Tel.: +(734)-232-0445; Fax: +734-615-8739; 2Center for Research on Ethnicity, Culture and Health, School of Public Health, University of Michigan, Ann Arbor, MI 48109-2700, USA

**Keywords:** socioeconomic status, education, ethnicity, racial and ethnic groups, Blacks, African Americans

## Abstract

Background. According to the Minorities’ Diminished Returns (MDR) theory, the health returns of socioeconomic status (SES) are systemically smaller for Blacks compared to Whites. Less is known, however, about trans-generational aspects of such diminished gains. For example, the differential impact of parental educational attainment on differences in mental well-being between White versus Black college students remains unknown. Aims. With a national scope, this study explored racial differences in the effect of parental educational attainment on the mental well-being of college students in the United States. Methods. The Healthy Mind Study (HMS), 2016–2017, is a national telephone survey that included 41,898 college students. The sample was composed of Whites (*n* = 38,544; 92.0%) and Blacks (*n* = 3354; 8.0%). The independent variable was highest parental educational attainment. The dependent variable was mental well-being (mentally healthy days in the past month) which was measured using a single item. Age and gender were covariates. Race was the moderator. Logistic regression was used for data analysis. Results. In the pooled sample, high parental educational attainment was associated with better mental wellbeing, independent of race, age, and gender. Race, however, showed a significant interaction with parental educational attainment on students’ mental wellbeing, indicating a smaller effect of parent education on the mental wellbeing of Black compared to White college students. Conclusions. The returns of parental educational attainment in terms of mental well-being is smaller for Black college students compared to their White counterparts. To minimize the diminished returns of parental education in Black families, policies should go beyond equalizing SES and reduce the structural barriers that are common in the lives of Black families. Policies should also aim to reduce extra costs of upward social mobility, prevent discrimination, and enhance the quality of education for Blacks. As the mechanisms that are involved in MDR are multi-level, multi-level solutions are needed to minimize the racial gaps in gaining mental health benefits from higher socioeconomic levels.

## 1. Background

The socioeconomic status (SES) of the family and the individual are both strong determinants of health [1,2,3,4,5,6]. SES resources due to education and employment of self and parents is protective against a wide range of poor health outcomes [7,8]. Families that have a low SES experience higher levels of undesired health outcomes across several health domains [9]. Family SES (e.g., parental educational attainment) is a salient factor that shapes offspring’s health [10,11,12]. Low family SES is, therefore, one of the mechanisms behind racial health disparities [10,13].

Individual and family SES, however, may not equally protect various sociodemographic groups against undesired health outcomes [14]. Majority and minority groups differently gain health from their access to a wide range of SES resources [15]. In the US, for example, the magnitude and the mechanism by which SES protects the health and well-being of populations widely varies across sub-populations [16], which is at least in part because of racism and discrimination [17,18,19,20]. Due to the differential treatment of sub-populations by society, social groups differ in their chance to mobilize their SES resources, navigate the system, and turn their potentials into tangible outcomes [21,22]. As minority group members tend to pay additional psychosocial costs for upward social mobility than the majority group [23,24], they may require more effort to climb the social ladder, which takes its toll [25,26,27]. Given the history of slavery, segregation, and structural racism, the labor market has particular preferences toward majority over minority group members and, as a result, education does not result in equal income and employment opportunities for minorities [14,15]. Lower education quality in poor urban communities also results in a smaller health gain of education for minorities [28]. Finally, minorities more frequently experience discrimination [21,29], which is a risk factor for a wide range of health problems [30,31,32]. At the same time, discrimination reduces the protective effects of SES resources [21,33]. All of these processes may be involved in explaining the diminished health gain of SES for minorities compared to majority group members. 

A number of recent studies show that education attainment translates to lower health gain for Blacks than Whites [14,15]. Protective effects of education on drinking alcohol [34], smoking [35], depression [36], suicide [37], chronic medical disease [36], obesity [38], and all-cause mortality [28,39,40,41] are shown to be smaller for Blacks than Whites. Although most of this evidence is on the effects of the individual or family SES of index individuals, a recent study showed that similar patterns may exist for transgenerational effects of SES [38]. Multiple studies used the Fragile Families and Child Wellbeing Study (FFCWS) data that had 15 years of follow up data of Black and White families. These studies revealed interactions between race and parental SES on subsequent youth body mass index (BMI) [38], impulsivity [42], and self-rated health [43], indicating smaller health gains of parental SES for Blacks compared to Whites.

When it comes to mental health outcomes, the story becomes even more complex. For some mental health outcomes, not only does SES not show a protective effect against the health of Blacks, high SES may increase the risk of poor mental health outcomes. For instance, among male Black youth, higher income increased the risk of Major Depressive Disorder (MDD) [44]. Among Black women, high education increased the risk of suicidal ideation [37], and high education at baseline predicted an increase in subsequent depressive symptoms for Black men [36]. Higher income was also shown to increase the risk of MDD in Black men [45,46].

To better understand the racial differences in transgenerational effects of SES on the mental well-being of college students, this study used a national sample to compare Black and White college students on the effects of parental SES (i.e., highest education of parents in the household) on the mental well-being of college students in the United States.

## 2. Methods

### 2.1. Design and Setting

This cross-sectional study used data from the Healthy Mind Study (HMS), an online mental health survey of college students in the United States. The HMS is a web-based survey that examines the mental health and well-being of undergraduate and graduate students in colleges in the United States. The survey gathers a wide range of data on demographic factors, socioeconomic status, daily experiences, mental health, stigma, substance use, and service use [47,48,49,50]. Since 2007, HMS has been conducted in 150 colleges with over 175,000 respondents.

### 2.2. Ethics

The HMS protocol has been approved by the Health Sciences and Behavioral Sciences Institutional Review Board (IRB) at the University of Michigan (UM), Ann Arbor. The study also received a Certificate of Confidentiality from the National Institutes of Health (NIH). All of the subjects gave their written informed consent for inclusion before they participated in the study. The study fully protected the privacy of its participants.

### 2.3. Sampling and Participants

Participating colleges provided the HMS team with a random sample of enrolled adult (over the age of 18) students. Large colleges provided a sample of 4000 enrolled students. Smaller colleges provided a census of all of their enrolled students. Schools with graduate students included both undergraduates and graduate students in their sample pool.

The HMS is a web-based survey. Students were invited to participate in the survey via emails. Participants also received reminders to participate in the survey. Non-responders were contacted by being sent up to three reminders that were spaced by two to four days each. Each invitation contained a URL that directed the students to the survey website.

### 2.4. Data Collection

The HMS is a web-based survey that uses three standard survey modules on all campuses: (1) Demographics; (2) Mental Health; and (3) Mental Health Service Use. The current analysis included the following variables: race, gender, age, SES (parental education), and mental well-being. 

*Race* In the current study, race/ethnicity was self-identified. Race included Blacks/African Americans and Whites. 

*Parental Education (SES).* In this study, we used highest level of parental education as the independent variable. A single item was used to measure parental educational attainment. “What is the highest level of education completed by your parents or stepparents?” (1) 8th grade or less; (2) 9th–12th grade (no high school degree); (3) high school degree; (4) Some college education (no college degree); (5) associate’s degree; (6) bachelor’s degree; and (7) graduate degree. Highest level of parents’ education was calculated by comparing the education level of both parents (if available). Parental education was operationalized as an interval measure ranging from 1 to 7, with higher scores indicating higher parental educational attainment. 

*Mental Well-Being.* Borrowed from the Centers for Disease Control and Prevention (CDC) Healthy Days questions that are used in the CDC’s Health-related quality of life (HRQOL) measure, we used the following item to measure mental well-being: “In the past 4 weeks, how many days have you felt that emotional or mental difficulties have hurt your academic performance?” Responses included (1) None; (2) 1–2 days; (3) 3–5 days; and (4) 6 or more days. This item has been used in the Behavioral Risk Factor Surveillance System (BRFSS), the National Health and Nutrition Examination Survey (NHANES), the Medicare Health Outcome Survey (HOS), the Healthcare Effectiveness Data and Information Set (HEDIS), and the National Commission for Quality Assurance (NCQA) [51].

### 2.5. Data Analysis

We used the SPSS 20.0 (IBM Inc., Armonk, NY, USA) statistical package for our data analysis. For descriptive purposes, we reported frequency (%) and mean with standard deviations (SD). For bivariate analysis, the Pearson correlation test was used. For multivariable analysis, we ran a series of logistic regression models. In all of the models, mental well-being was the dependent variable, family SES (parental educational attainment) was the main independent variable of interest, and age and gender were the study covariates.

First, we ran two logistic multivariable regressions in the overall sample. The first model only included the main effects of parental SES, race, and study covariates. The second model also included a race × parental educational attainment interaction term. Then, we ran stratified models in each race. Adjusted (unstandardized) regression coefficients (b), 95% Confidence Interval (CI), and p values are reported. 

## 3. Results

### 3.1. Descriptive Results

This analysis included 41,898 college students who were at least 18 years old. The sample was mostly females (*n* = 28,967; 69.22%) and only 30.78% were males (*n* = 12,883). Table 1 summarizes the descriptive statistics for the pooled sample, as well as based on race (Table 1).

### 3.2. Bivariate Associations

As Table 1 shows, Blacks and Whites differed in age, gender, SES, and mental well-being. Overall, Blacks were older, had more females, reported lower education of their parents, and had a higher prevalence of poor mental well-being than Whites (Table 1).

Table 2 provides the results of the bivariate associations in the pooled sample. In the pooled sample, age and gender were associated with mental well-being (Table 2).

### 3.3. Logistic Regressions in the Overall Sample

Table 3 summarizes the results of the two logistic regressions in the overall sample, one without the interaction, and one with the race by SES interaction. *Model 1* showed that in the pooled sample, parental education had a significant association with odds of having poor mental well-being. *Model 2* showed a significant interaction between race and parental education on mental well-being, suggesting a weaker protective effect of parental education on the odds of poor mental well-being for Blacks compared to Whites (Table 3).

### 3.4. Race-Specific Logistic Regressions

Table 4 summarizes the results of the two logistic regression models specific to race. *Model 3* showed a significant association between parental education and odds of poor mental well-being for Whites. *Model 4*, however, did not show any association between parental education and odds of poor mental well-being for Blacks (Table 4).

## 4. Discussion

We found an overall protective effect of parental SES (i.e., parental education) in reducing the odds of poor mental well-being. We also found some Black-White variation in the protective effect of parental educational attainment on the mental well-being of the college students. Only White but not Black college students, showed mental well-being gains from their parental education. 

These findings are in harmony with previous studies that have documented diminished returns of SES indicators for Blacks across age groups, populations, and outcomes [14,15]. However, these studies have been mostly limited to one generation. There are only a few multi-generational studies documenting Blacks’ diminished returns. In a recent study using national data of children, Black children’s risk of being overweight did not decline with an increase in family SES. This pattern was different from White children’s risk of being overweight which was a function of family SES [52]. In another study that used the 15-year follow data from the FFCWS study, associations were found between parental SES and marital status at birth and youth BMI at age 15 for Whites, however not for Blacks [38].

As the current study and previous studies show, the very same SES indicators seem to be less protective against poor health outcomes for the economically disadvantaged (minority) group than the socially privileged (majority) group. The current finding adds a multigenerational aspect to what is already shown for children and youth [38,46], adults [36,37], and older adults [16,53]. Thus, both the same person’s SES and also parent’s SES better translate in health gains for Whites than Blacks. Although the exact underlying mechanism behind these differential gains is still unknown, they start early in life and transition from one generation to another [38,46]. Differential transgenerational effects of SES on health may be one mechanism for differential effects of own SES on health. 

As we have suggested elsewhere, the findings reported here, and in other studies on Blacks’ diminished returns, do not indicate Blacks’ inability or low efficiency in translating their SES resources to tangible health outcomes, which would be blaming the victim [54]. It is not the group itself, however it is how society treats the group that shapes how easy a social group can mobilize economic resources such as education. Given the history of slavery and existing racism that affects almost all aspects of American social structure and function, SES resources better serve Whites than Blacks [14,15]. Blacks and other minorities use a considerable amount of their energy to deal with stressors and unjust treatment which are both disproportionately prevalent in their lives. Thus, it is the high prevalence of societal and structural barriers in the lives of Blacks that hinder them from gaining tangible outcomes from their SES resources. 

Thus, it is not the Blacks themselves, however it is the American social system that is to blame for these differential and unequal effects. The American social and economic system has historically failed Blacks by charging them more and forcing them to pay extra costs for social mobility. Social mobility is an uphill battle for minorities, particularly Blacks [23,24]. The way American society operates is to ensure that it consistently maximizes the gains of the privileged group, even if it comes with a cost to the minority groups [14,15]. This is why the Black-White economic gap is rising in the United States. 

Despite high aspirations, high SES Black families face more barriers that reduce their success. One example is the discrimination that reduces the expected health gain that is associated with higher SES [36]. In a race–and–color–aware society who treats groups based on their color rather than their potentials, high aspiration may even operate as a risk factor for Blacks. Similar results are shown for the effects of self-efficacy, sense of control over life, and mastery, as such traits may become harmful in environments that are very difficult to control [55,56]. In support of this theory, research has shown that psychological constructs such as self-efficacy, sense of control over life, and mastery protect Whites, however not Blacks [57,58].

Thus, SES gains are smaller for Blacks, which is a pattern that is not limited to any age group [14,15]. For example, educational attainment better changed the drinking habits of White than Black older adults [53] and education had a protective effect against obesity, physical inactivity, and poor sleep quality for Whites, however not for Blacks who were older than the age of 50 [16]. Similarly, educational attainment [28], employment status [59], neighborhood quality [60], and number of social contacts [61] had a smaller effect on the life expectancy for Blacks than for Whites. All of these findings were in line with the minorities’ diminished returns [16,28,37,62].

## 5. Limitations

Our study had a number of limitations. First and foremost, due to the cross-sectional design, this study is unable to conclude causal inferences. Reverse causation, however, can be ruled out here, as it is very unlikely that poor mental wellbeing of college students resulted in low education of parents. Still, these findings should be replicated in longitudinal studies with multiple observations [63,64,65,66]. Second, this study only includes Blacks and Whites, and other racial and ethnic groups were not included. Third, all of the potential confounders were not controlled in this study. In addition to confounders, future studies should also consider other potential effect modifiers such as ethnicity, region, and college characteristics. There is a need to replicate the current finding across other socially marginalized groups such as Native Americans, Latinos, sexual minorities, and immigrants. Fourth, our SES indicators were not a comprehensive list. Other SES indicators such as family type, household size, employment, wealth, and income should be investigated. Last but not least, we did not study the exact mechanism behind Blacks’ diminished returns. We only documented them. Future research should explore whether stress, discrimination, or behaviors influence these effects. Despite these methodological limitations, this study was one of very few cross-generational studies on Blacks’ diminished returns.

## 6. Conclusions

To conclude, Black college students gain less mental well-being from their parental educational attainment, possibly due to racism, discrimination, and societal barriers in their daily lives. Multi-level policies should address multi-level barriers in the lives of Blacks. Policies need to go beyond equalizing SES resources across racial groups. True equity cannot be achieved unless all racial groups are treated fairly so that they can have the same chance to gain tangible outcomes from their resources.

## Figures and Tables

**Table 1 brainsci-08-00193-t001:** Descriptive statistics overall and by race.

	**All (*n* = 41,898)**	**Whites (*n* = 38,544)**	**Blacks (*n* = 3354)**
	***n*(%)**	***n*(%)**	***n*(%)**
**Gender ***			
Male	28,967(69.22)	26,507(68.85)	891(26.59)
Female	12,883(30.78)	11,992(31.15)	2460(73.41)
**Mental Well-being ***			
High	33,736(82.16)	31,104(82.27)	2632(80.98)
Low	7323(17.84)	6705(17.73)	618(19.02)
	**Mean(CI)**	**Mean(CI)**	**Mean(CI)**
**Age (Year) ***	23.70(7.15)	23.53(6.89)	25.73(9.39)
**Education ***	5.64(1.42)	5.70(1.38)	4.96(1.68)

* *p* < 0.05.

**Table 2 brainsci-08-00193-t002:** Correlation matrix in the pooled sample and by race.

	1	2	3	4	5
**All (*n* = 41,898)**					
1 Race (Blacks)	1	0.084 **	−0.027 **	−0.140 **	0.009
2 Age (Year)		1	0.013 *	−0.209 **	−0.023 **
3 Gender (Females)			1	0.055 **	−0.035 **
4 Education				1	−0.027 **
5 Mental Well-being					1
**Whites (*n* = 38,544)**					
2 Age (Year)	-	1	0.014 **	−0.187 **	−0.019 **
3 Gender (Females)	-		1	0.060 **	−0.032 **
4 Education	-			1	−0.032 **
5 Mental Well-being	-				1
**Blacks (*n* = 3354)**					
2 Age (Year)	-	1	0.023	−0.291 **	−0.069 **
3 Gender (Females)	-		1	−0.040 *	−0.064 **
4 Education	-			1	0.031
5 Mental Well-being	-				1

* *p* < 0.05; ** *p* < 0.01

**Table 3 brainsci-08-00193-t003:** Summary of linear regression models in the pooled sample.

	Pooled Sample (*n* = 41,898)			
	OR	(95% CI)	OR	(95% CI)
	*Model 1* *Main Effects*		*Model 2* *Main Effects + Interaction*	

Race (Blacks)	0.94	0.86–1.03	1.51 **	1.11–2.05
Age (Year)	0.99 ***	0.98–0.99	0.99 ***	0.99–0.99
Gender (Females)	1.21 ***	1.14–1.28	1.20 ***	1.14–1.27
Parental Education	0.95 ***	0.93–0.96	0.94 ***	0.92–0.95
Parental Education * Race	-	-	1.10 ***	1.04–1.16
Intercept	0.36 ***	-	0.23 ***	-

Outcome: poor mental well-being, OR: Odds Ratio; CI: Confidence Interval; * *p* < 0.05, ** *p* < 0.01, *** *p* < 0.001.

**Table 4 brainsci-08-00193-t004:** Summary of two logistic regression models specific to race.

	Whites (*n* = 38,544)		Blacks (*n* = 3354)	
	OR	(95%CI)	OR	(95%CI)
Race (Blacks)				
Age (Year)	0.99 ***	0.99–0.99	0.98 **	0.97–0.99
Gender (Females)	1.19 ***	1.12–1.26	1.44 ***	1.16–1.79
Parental Education	0.94 ***	0.92–0.96	1.02	0.96–1.08
Intercept	0.35 ***		0.26 ***	

Outcome: poor mental well-being, CI: Confidence Interval; ** *p* < 0.01, *** *p* < 0.001.

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
