# Peer review of "Parental Educational Attainment and Mental Well-Being of College Students: Diminished Returns of Blacks"

_brainsci, 2018, doi:10.3390/brainsci8110193_

Reviewer 1 Report

This study belongs to a series of works documenting the evidence for diminished returns effects, it is well written with well documented references to relevant studies. I have some minor cosmetic suggestions for author(s):

abstract
Independent variable was highest parental education attainment. Dependent variable was mental well-being
->
The independent variable was highest parental education attainment. The dependent variable was mental well-being
As the involved mechanisms are multi-level, multi-level solutions
As the mechanisms involved in MDR are multi-level, multi-level solutions
p.1:
Availability of SES due to education, and employment of self and parents are protective
Availability of SES due to education, and employment of self and parents is protective
Families that have low SES experience higher levels of undesired health outcomes across domains
Families that have low SES experience higher levels of undesired health outcomes across several domains
Poor family SES is one of the mechanisms behind racial health disparities
Poor family SES is therefore one of the mechanisms behind racial health disparities
Majority and minority groups differently gain health from
Majority and minority groups differently gain health from

p.2

In U.S. for example, the magnitude and the

In the U.S. for example, the magnitude and the
This is mainly due to racism and discrimination [].
[I would delete]
minority groups may turn to high effort copings to climb the social ladder, which takes it’s toll

minority groups may require more effort to climb the social ladder, which takes its toll
[several references might need entered:]
smoking [], obesity [], impulsivity [], and self-rated health []
For instance, among male Black youth, high income increased the risk of Major Depressive Disorder (MDD)

For instance, among male Black youth, higher income increased the risk of Major Depressive Disorder (MDD)
Among Black women, high education increased the risk of suicidal ideation

Among Black women, higher education increased the risk of suicidal ideation
High income is also shown to increase risk of MDD in Black men [35,36].

Higher income is also shown to increase risk of MDD in Black men [35,36].
All participants provide infomed consent,

All participants provide informed consent,

p.3
For descriptive purposes, we used frequency (%) and mean with standard deviations (SD).

For descriptive purposes, we ureport frequency (%) and mean with standard deviations (SD).
Adjusted (unstandardized) regression coefficients (b), 95 % Confidence Interval (CI), and p values were reported.

Adjusted (unstandardized) regression coefficients (b), 95 % Confidence Interval (CI), and p values are reported.
 p. 4
Overall, Blacks were older, were more female, and reported lower education of their parents and had a higher prevalence of poor mental well-being.

Overall, Blacks were older, had more females, reported lower education of their parents and had a higher prevalence of poor mental well-being than Whites.
p.5
weaker protective effect of parental education on odds of poor mental well-being for Blacks

weaker protective effect of parental education on the odds of poor mental well-being for Blacks
This pattern was different from White children’s risk of overweight

This pattern was different from White children’s risk of being overweight

p.6
Thus both the same person SES and also parent SES

Thus both a person's SES and also their parents' SES
diminished return do not indicate Blacks’ inability or low efficiently in translating their available SES

diminished return do not indicate Blacks’ inability or low efficiency in translating their available SES
American system has historically failed Blacks

The American system has historically failed Blacks
Social mobility is fighting an uphill battle for minorities particularly Blacks

Social mobility is an uphill battle for minorities, particularly Blacks
The way American society operates is to ensure it constantly maximizes the gain of the privileged group,
The way American society operates is to ensure it constantly maximizes the gain of the privileged groups,
One example is the discrimination that reduces the expected health gain associated with high SES

One example is the discrimination that reduces the expected health gain associated with higher SES

p.7
However, it is very unlikely that poor mental wellbeing of college students result in low education of parents.

Reverse causation for one, however, can be ruled out, as it is very unlikely that poor mental wellbeing of college students result in low education of parents.

p.8
Despite these methodological limitations, this study was one of very few cross-generational studies on Blacks’ diminished return.

Despite these methodological limitations, this study was one of very few cross-generational studies on Blacks’ diminished returns.

p.9
A true equity cannot be achieved unless all racial groups are treated fairly so they can have the same chance to gain tangible outcomes from their resources.

True equity cannot be achieved unless all racial groups are treated fairly so they can have the same chance to gain tangible outcomes from their resources.

Author Response

Thank you for very helpful comments.

Here are the changes (all the changes are in red):

abstract
Independent variable was highest parental education attainment. Dependent variable was mental well-being -> The independent variable was highest parental education attainment. The dependent variable was mental well-being
As the involved mechanisms are multi-level, multi-level solutions -> As the mechanisms involved in MDR are multi-level, multi-level solutions
p.1
Availability of SES due to education, and employment of self and parents are protective
Availability of SES due to education, and employment of self and parents is protective
Families that have low SES experience higher levels of undesired health outcomes across domains -> Families that have low SES experience higher levels of undesired health outcomes across several domains
Poor family SES is one of the mechanisms behind racial health disparities -> Poor family SES is therefore one of the mechanisms behind racial health disparities
Majority and minority groups differently gain health from -> Majority and minority groups differently gain health from

 p.2

In U.S. for example, the magnitude and the -> In the U.S. for example, the magnitude and the

rephrased the phrase “This is mainly due to racism and discrimination [].
minority groups may turn to high effort copings to climb the social ladder, which takes it’s toll -> minority groups may require more effort to climb the social ladder, which takes its toll
Added references to :   “smoking [], obesity [], impulsivity [], and self-rated health []”
For instance, among male Black youth, high income increased the risk of Major Depressive  Disorder (MDD) -> For instance, among male Black youth, higher income increased the risk of Major Depressive Disorder (MDD)
Among Black women, high education increased the risk of suicidal ideation -> Among Black women, higher education increased the risk of suicidal ideation
High income is also shown to increase risk of MDD in Black men [35,36].->Higher income is also shown to increase risk of MDD in Black men [35,36].
All participants provide infomed consent, -> All participants provide informed consent,

p.3
For descriptive purposes, we used frequency (%) and mean with standard deviations (SD). -> For descriptive purposes, we report frequency (%) and mean with standard deviations (SD).
Adjusted (unstandardized) regression coefficients (b), 95 % Confidence Interval (CI), and p values were reported. -> Adjusted (unstandardized) regression coefficients (b), 95 % Confidence Interval (CI), and p values are reported.
p. 4
Overall, Blacks were older, were more female, and reported lower education of their parents and had a higher prevalence of poor mental well-being. -> Overall, Blacks were older, had more females, reported lower education of their parents and had a higher prevalence of poor mental well-being than Whites.
p.5
weaker protective effect of parental education on odds of poor mental well-being for Blacks -> weaker protective effect of parental education on the odds of poor mental well-being for Blacks
This pattern was different from White children’s risk of overweight -> This pattern was different from White children’s risk of being overweight

p.6
Thus both the same person SES and also parent SES -> Thus both a person's SES and also their parents' SES
diminished return do not indicate Blacks’ inability or low efficiently in translating their available SES -> diminished return do not indicate Blacks’ inability or low efficiency in translating their available SES
American system has historically failed Blacks -> The American system has historically failed Blacks
Social mobility is fighting an uphill battle for minorities particularly Blacks -> Social mobility is an uphill battle for minorities, particularly Blacks
The way American society operates is to ensure it constantly maximizes the gain of the privileged group,
The way American society operates is to ensure it constantly maximizes the gain of the privileged groups,
One example is the discrimination that reduces the expected health gain associated with high SES
One example is the discrimination that reduces the expected health gain associated with higher SES

p.7
However, it is very unlikely that poor mental wellbeing of college students result in low education of parents. -> Reverse causation for one, however, can be ruled out, as it is very unlikely that poor mental wellbeing of college students result in low education of parents.

p.8
Despite these methodological limitations, this study was one of very few cross-generational studies on Blacks’ diminished return. -> Despite these methodological limitations, this study was one of very few cross-generational studies on Blacks’ diminished returns.

p.9
A true equity cannot be achieved unless all racial groups are treated fairly so they can have the same chance to gain tangible outcomes from their resources. -> True equity cannot be achieved unless all racial groups are treated fairly so they can have the same chance to gain tangible outcomes from their resources

Reviewer 2 Report

Overall this is very important work and worthy of publication.  Examining the differential impact of SES status on health outcomes for different groups is important for distinguishing how social determinants influence health. However, there are many areas that need to be refined to make it a high quality publication.

MECHANICS

The entire paper requires editing. It is laden with errors in grammar, word choice, spelling errors and awkward wording. Examples are below.

Under Abstract:

1.       Some thought can be made more concise for parsimony. For example in the abstract the author states, “We still do not know whether the effect of parental education attainment on mental well-being differences for White and Black college students or not.” This can be changed to, “The differential impact of parental educational attainment on differences in mental well-being between White versus Black college students remains unknown.”

2.       Under results some letter missing. References to education attainment should read educational attainment.

3.       Conclusions: Some thoughts are incomplete and imprecise, awkward wording.

4.       last sentence under the abstract should read “gaining mental health benefits from higher socioeconomic levels.

Under Background:

1.       First sentence: The definite article “The” is often left out.  “The” socioeconomic status of the family, not simply socioeconomic status.

2.       2nd sentence. We don’t refer to SES as available. Just say, Socioeconomic factors such as education and employment status of parents are protective…….

3.       3rd sentence. The definite article “a” is missing in front of low ..should read “a” low SES and at the end of that sentence define “domains” …what domains are you referring to?

4.       4th sentence. “offspring should have an apostrophe then s since health is owned by the offspring. Should read, the “offspring’s” health.

5.       5th sentence. Refer to SES as “low” not as “poor”. So this should read, Low SES is one of the mechanisms behind racial disparities……

6.       2nd paragraphs 1st sentence: Type…. “again” should read “against”

7.       2nd sentence 2nd paragraph: Poor word choice. “differently gain health” that phrase is unclear and incorrect. We do not “gain health” How about saying “ Take this sentence out because it is redundant since the next sentence already captures your thought and is more accurately phrased, when it states, “In the U.S. for example……..

8.       The entire 2nd paragraph under background has very poor sentence structure, mechanical errors, inappropriate word insertions and typos. These aforementioned errors occur extensively throughout the paper. All examples can’t be cited. The entire paper need editing.

9.       Discussion: “American system has historically failed Blacks by charging them more and forcing them pay extra costs for social mobility.” This sentence has missing words and typos.

CONTENT:

Background:

This study is lacking in a theory. An abundance of research findings are presented without the underlying theoretical base to serve as a framework for the analysis. Perhaps a discussion of the theoretical interactions involved in transgenerational transmission theories may apply. References were made about diminished returns yet this is not explained.

Methods

Need reliability and validity data of the Mental Well-Being survey borrowed from the CDC. Add the reliability alpha coefficient in this sample.

Limitations: One statement is opposite your main point. You state. “however, it is very unlikely that poor mental well being of college students results in low education of parents.” “This should state it is unlikely that the low educational level of parents causes poor mental well being of college students.”

Author Response

Thank you very much for the very helpful comments and edit.

Here is each comment / changes. All the changes in the text are in red.

The entire paper requires editing. It is laden with errors in grammar, word choice, spelling errors and awkward wording. Examples are below.

We edited the paper.

Under Abstract:

1.       Some thought can be made more concise for parsimony. For example in the abstract the author states, “We still do not know whether the effect of parental education attainment on mental well-being differences for White and Black college students or not.” This can be changed to, “The differential impact of parental educational attainment on differences in mental well-being between White versus Black college students remains unknown.”

2.       Under results some letter missing. References to education attainment should read educational attainment.

Changed/ Corrected. 

3.       Conclusions: Some thoughts are incomplete and imprecise, awkward wording.

 Changed/ Corrected.

4.       last sentence under the abstract should read “gaining mental health benefits from higher socioeconomic levels.

 Changed/ Corrected.

Under Background:

 1.       First sentence: The definite article “The” is often left out.  “The” socioeconomic status of the family, not simply socioeconomic status.

 Changed/ Corrected.

2.       2nd sentence. We don’t refer to SES as available. Just say, Socioeconomic factors such as education and employment status of parents are protective…….

 Changed/ Corrected.

3.       3rd sentence. The definite article “a” is missing in front of low ..should read “a” low SES and at the end of that sentence define “domains” …what domains are you referring to?

 Changed/ Corrected.

4.       4th sentence. “offspring should have an apostrophe then s since health is owned by the offspring. Should read, the “offspring’s” health.

 Changed/ Corrected.

5.       5th sentence. Refer to SES as “low” not as “poor”. So this should read, Low SES is one of the mechanisms behind racial disparities……

 Changed/ Corrected.

6.       2nd paragraphs 1st sentence: Type…. “again” should read “against”

 Changed/ Corrected.

7.       2nd sentence 2nd paragraph: Poor word choice. “differently gain health” that phrase is unclear and incorrect. We do not “gain health” How about saying “ Take this sentence out because it is redundant since the next sentence already captures your thought and is more accurately phrased, when it states, “In the U.S. for example……..

 Changed/ Corrected. This sentence is revised.

8.       The entire 2nd paragraph under background has very poor sentence structure, mechanical errors, inappropriate word insertions and typos. These aforementioned errors occur extensively throughout the paper. All examples can’t be cited. The entire paper need editing.

 Changed/ Corrected. This paragraph is revised.

9.       Discussion: “American system has historically failed Blacks by charging them more and forcing them pay extra costs for social mobility.” This sentence has missing words and typos.

 Changed/ Corrected.

CONTENT:

Background:

This study is lacking in a theory. An abundance of research findings are presented without the underlying theoretical base to serve as a framework for the analysis. Perhaps a discussion of the theoretical interactions involved in transgenerational transmission theories may apply. References were made about diminished returns yet this is not explained.

The MDR is a theory. The whole paper is about minorities diminished return theory. Theory papers are also cited.

Methods

Need reliability and validity data of the Mental Well-Being survey borrowed from the CDC. Add the reliability alpha coefficient in this sample.

Our outcome is a single item measure, and single item measures do not have reliability and reliability alpha coefficient. So, there is no way to give reliability of this item / measure.

Limitations: One statement is opposite your main point. You state. “however, it is very unlikely that poor mental well being of college students results in low education of parents.” “This should state it is unlikely that the low educational level of parents causes poor mental well being of college students.”

No. Here we are talking about reverse causality! So, it is not x on y. It is the chance that y impacts x. So, it is correct..